# Knowledge, attitude and practice towards kangaroo mother care among postnatal women in Ethiopia: Systematic review and meta-analysis

**Natnael Atnafu Gebeyehu**[1]*, **Kelemu Abebe Gelaw**[1], **Gedion Asnake Azeze**[1], **Biruk Adie Admass**[2], **Eyasu Alem Lake**[3], **Getachew Asmare Adela**[4]

1 School of Midwifery, College of Health Science and Medicine, Wolaita Sodo University, Wolaita Sodo, Ethiopia, 2 Department of Anesthesia, College of Health Science and Medicine, University of Gondar, Gondar, Ethiopia, 3 School of Nursing, College of Health Science and Medicine, Wolaita Sodo University, Wolaita Sodo, Ethiopia, 4 School of Public Health, College of Health Science and Medicine, Wolaita Sodo University, Wolaita Sodo, Ethiopia

* jossyatnafu2020@gmail.com

**Data Availability Statement:** All relevant data are within the Manuscript and its Supporting Information files.

## Abstract

### Background

Kangaroo mother care is a key procedure in reducing neonatal mortality and morbidity associated with preterm birth. In Ethiopia, neonatal death remains a serious problem, and this study aims to determine the prevalence of the knowledge, attitudes and practice of kangaroo mother care among Ethiopia women.

### Methods

PubMed, Web of Science, Google Scholar, EMBASE and the Ethiopian University online library were searched. Data were extracted using Microsoft Excel and analysed using STATA statistical software (v. 11). Publication bias was checked by forest plot, Begg's rank test and Egger's regression test. To look for heterogeneity, $I^2$ were computed and an overall estimated analysis carried out. Subgroup analysis was done by region, study setting, publication, gestational age, birth weight and component of kangaroo care. The Joanna Briggs Institute risk of bias assessment tool was used. We carried out a leave one out sensitivity analysis.

### Results

Out of 273 articles retrieved, 16 studies met the eligibility criteria and are thus included in this study. Those 16 studies had a total of 12,345 respondents who reported kangaroo mother practice, with five (comprising 1,232 participants combined) reporting that both knowledge and attitude were used to determine the overall estimation. The pooled estimates of good knowledge, positive attitude and poor practice of kangaroo mother care were found to be 64.62% (95% CI: 47.15%–82.09%; $I^2$ = 97.8%), 61.55% (49.73%–73.38%; $I^2$ =

**Funding:** The authors received no specific funding for this work.

**Competing interests:** The authors declared that they have no competing interests

94.8%) and 45.7% (95% CI: 37.23%–54.09%; $I^2$ = 98.5%), respectively. This study is limited to postnatal women and does not take account their domestic partners or health providers.

## Conclusion

The findings revealed significant gaps in the knowledge, attitudes and practice of kangaroo mother care in Ethiopia when compared with other developing countries. Therefore, kangaroo mother care training to women, along with further studies on domestic partners and health providers.

## Introduction

Low birth weight and preterm birth represent the two major public health challenges during the neonatal period [1]. During this period, 36% of deaths occur on the day of birth, while 73% occur within the first week of life [2]. In addition, More than 80% of neonatal deaths occur in low birth weight neonates two-thirds of whom were are born prematurely [3]. Complications related to prematurity are the major cause of neonatal mortality [4]. Worldwide, 25 million babies are born a low birth weight annually while 15 million are born prematurely with 96% of these preterm babies occurring in developing countries [5, 6]. In Ethiopia, systematic review and meta-analysis studies have reported the pooled prevalence of preterm birth (to be 10.8%), and that of low birth weight (to be 17.3%) [7, 8].

Due to the prevalence of preterm births and low birth weight, as well as associated health care burden, comprehensive interventional procedures for primary prevention are required such as Kangaroo Mother Care [9]. According to the world health organization (WHO), kangaroo mother care involves an early, uninterrupted, and prolonged skin–to–skin contact between mother and the baby [10]. Although kangaroo mother care is recommended by World Health Organization, Baby Friendly Initiatives, United Nation International Children Emergency Fund, and American Academy of pediatrics, the prompt separation of the baby from the mother after birth remains a significant challenge [11–14].

Several studies have reported that kangaroo mother care to be a cost-effective intervention for reducing mortality and morbidity in preterm infants [15]. It has also been found to have a positive impact on maternal health in low, middle, and high-income countries [16–22]. In fact, studies have shown that kangaroo mother care reduced neonatal mortality [17, 23], sepsis [17, 23], hypothermia [17, 23], hypoglycemia [23], and length of hospital stay [17] when compared with conventional care approaches.

The findings of a number of systematic review studies have indicated that kangaroo mother care increased the success, initiation, and duration of breast feeding [24–29]. Moreover, it has also been found to improve maternal anxiety and stress [30], enhanced-cognitive and motor development [31], reduce the likelihood of hospital readmission [32, 33] and lowered the premature infant profile [34]. Another study reported that Kangaroo mother care improves the growth of low birth weight and preterm infants [35–41].

In Ethiopia, kangaroo mother care was first introduced in 1996 at Black Lion Hospital in Addis Ababa. Since then the service has been expanded to other health facilities and hospitals. In addition, the Federal Ministry of Health has integrated kangaroo mother care into National Strategy for Newborn and Child Survival, Health Sector Transformation Plan, and National Health Care Quality Strategy with the later focusing on ensuring that 80% of preterm babies to receive kangaroo mother care, although initiation currently remains low [42–44]. Moreover,

the neonatal mortality rate in Ethiopia is 30 per 1000 live births, which means that the problem remains significant [45]. By the end of 2030, Ethiopia aims to have lowered the neonatal mortality rate to 12 deaths per 1000 live births [46]. To help achieve this target, Kangaroo mother care is expected to play a significant role in anticipating neonatal hypothermia [11].

The prevalence of kangaroo mother care has been determined to range from 1% in Tanzania [47] to 96% in Denmark [14]. In Ethiopia, knowledge of kangaroo mother care has been found to range from 35.5% to 82.53% [48–52], attitude from 50% to 82.53% [49–53] and practice from 23% to 83% across the nation [48–63]. Given these variations, there is no overall estimate of the prevalence of kangaroo mother care based on representative national data in Ethiopia. Therefore, the present study sought to determine the pooled prevalence of knowledge, Attitude, and practice regarding of kangaroo mother care among postnatal women in Ethiopia. The aim was to provide fundamental data for policymakers, clinicians, and other stakeholders in order to help develop an appropriate strategies and interventions for the control and management of low-birth weight and preterm birth using kangaroo mother care.

## Methods

### Searching strategy

To obtain the data required for this study, we performed manual searched and also searched on PubMed, Web of Science, Google Scholar, EMBASE, and various grey literature data bases (Addis Ababa University, Ethiopia). More specifically, we used Keywords, medical Subject headings (MeSH) terms and Emtree terms to conduct the search. We applied search terms independently and/or in combination using "OR", "AND" or "NOT". In EMBASE we used Boolean operators and Emtree terms (a controlled vocabulary or standard words used to make searching easier) to identify relevant articles, whereas in Web of science we used synonyms, Boolean operators, key words, and topics. In Google Scholar and when performing manual searches, we used a combination of the above mentioned key terms/phrases (i.e. "Knowledge of kangaroo mother care", "attitude of kangaroo mother care", "kangaroo mother care", "low birth weight neonate", "preterm neonate", "newborn care practice", "kangaroo mother", "skin to skin care practice", "skin to skincare", "kangaroo mother care method, and Ethiopia) were used. The grey literature databases were searched via Ethiopian Universities Online repository library home. Our search strategy with regard to PubMed was as follows: (((((((((Knowledge [tw]) OR "Knowledge"[Mesh Terms]) AND (Attitude[tw] OR perception [tw])) OR "Attitude " [Mesh Terms]) AND Practice [tw]) OR "Practice" [Mesh Terms]) AND (Kangaroo mother care[tw] OR low birth weight neonate[tw]OR preterm neonate[tw] OR newborn care practice[tw])) OR ("Kangaroo mother care " [Mesh Terms] OR "Kangaroo mother " [Mesh Terms] OR "skin to skin care practice" [Mesh Terms] OR "skin to skin care"[Mesh Terms] OR" kangaroo mother care method" [Mesh Terms])) AND Ethiopia. The electronic literature search was performed from 30 May, 2021 to 30 June, 2021. All of the accessible studies that had been published in English from inception up to 30 June, 2021 were included in the present meta-analysis and systematic review.

In terms of reporting the findings of the literature search, The Preferred Reporting Items for Systematic Reviews and Meta-Analysis (PRISMA) guideline was used [64] (S1 File). This systematic review and meta-analysis study was not registered under Prospero, but we checked that any author has not registered it yet.

### Operational definitions

**Knowledge.** Those study participants who were responded to ≥50% of the knowledge-related questions were considered to have a good level of knowledge, while those who responded to less than 50% were considered to have a poor level of knowledge [48].

**Attitude.**  Those study participants who answered ≥50% of the mean value of the on attitude-related questions were considered to exhibit a positive attitude towards kangaroo mother care whereas those who answered below the mean value of the attitude-related questions were considered to exhibit having a negative attitude.

**Practice.**  Those study participants who responded to ≥50% of the practice-related questions were categorized as demonstrating good practice, whereas those respondents who answered <50% of the questions were considered to demonstrate poor practice.

## Eligibility criteria

All studies that reported the prevalence of knowledge, attitude and practice of kangaroo mother care, postnatal women as study participants, English language reporting, had full text available for search and took place in Ethiopia were included in this systematic review and meta-analysis. Those studies that reported duplicated sources, unrelated research, case reports, qualitative studies, and articles with no full text available attempts to contact the corresponding author via email were excluded this systematic review and meta-analysis.

## Study selection and data extraction

Three independent authors selected the candidate articles for the study, which were exported Endnote reference manager software to remove duplicate, and independently screened the titles and abstracts (NA, KA, and EA). Any disagreement was resolved through discussions lead by a third author. Data were extracted using a standardized data extraction format prepared in Microsoft Excel by four independent authors (NA, GA, BA and GA). Any disagreement that happened during extraction was resolved through discussion lead by the fourth author. The data automation tool was not used due to the absence of the paper form (manual data) in this study. The name of the first author, year of publication, study region, study setting, the prevalence of knowledge of kangaroo mother care, the attitudes towards kangaroo mother care, the practice of kangaroo mother care, sample size, gestational age, type of kangaroo mother care and birth weight were collected.

## Quality assessment

The two independent authors appraised the standard of the studies using the Joanna Briggs Institute (JBI) quality appraisal checklist [31]. Any disagreement was discussed and resolved by the authors. The critical analysis checklist has eight parameters with yes, no, unclear, and not an applicable options. The parameters involve the following questions:

1. Where were the criteria for inclusion in the sample clearly defined?

2. Were the study subjects and therefore the setting described in detail?

3. Was the exposure measured the result validly and reliably?

4. Were the main objective, standard criteria used for measurement of the event?

5. Were confounding factors identified?

6. Were strategies to affect confounding factors stated?

7. Were the results measured truly and dependably?, and (8) Was the statistical analysis suitable?. Studies were considered low risk when they scored 50% and above of the quality assessment indicators as reported in supplementary file (S2 File).

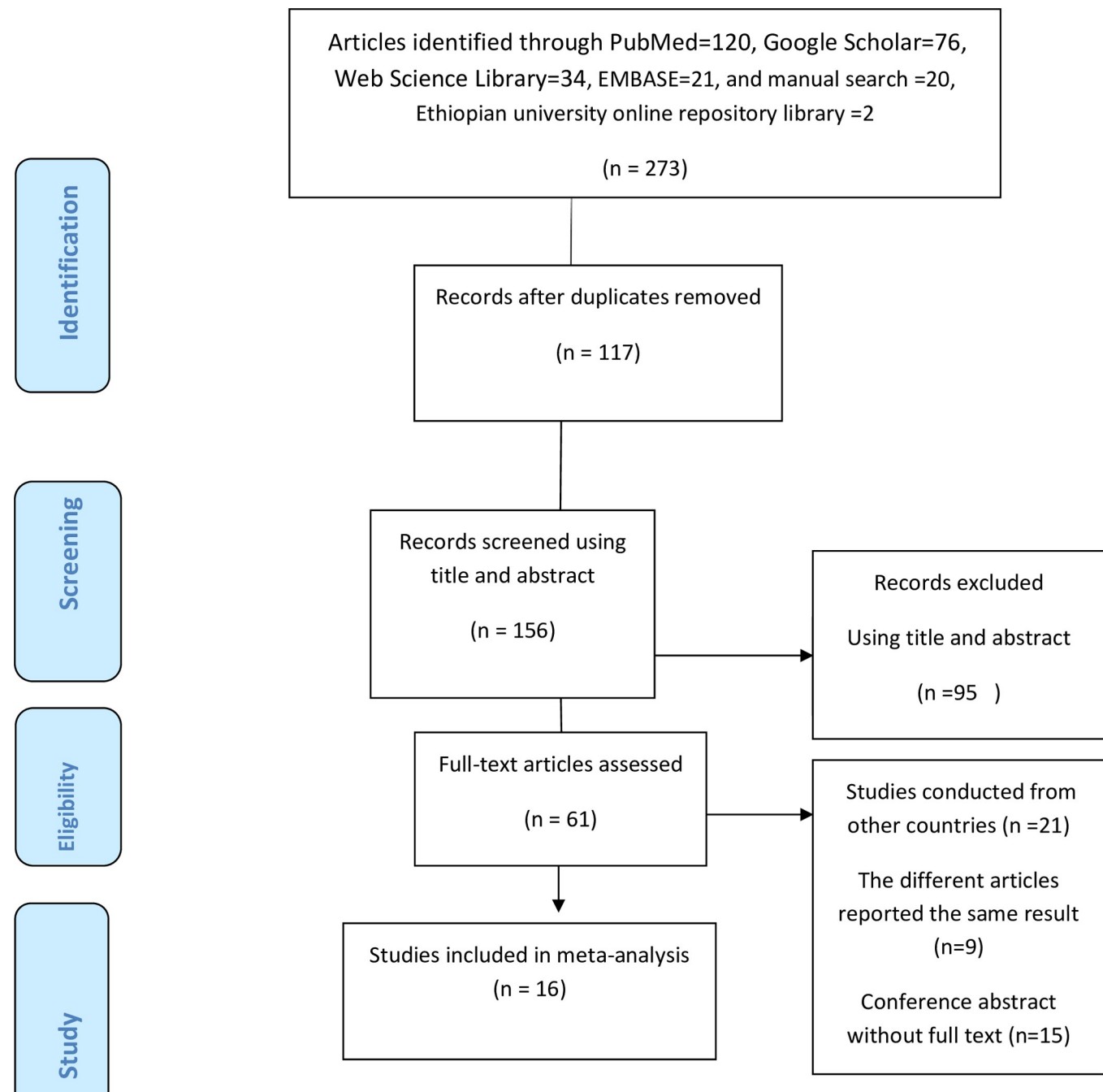

**Fig 1. A Prisma diagrammatic presentation used to show the selection of studies.** The inclusion criteria were variation of the title and abstracts, place of study (Ethiopia), presence of full abstract, and reporting different results. Studies were excluded if they criteria were duplicated source, unrelated research, case studies and qualitative studies.

**Risk of bias assessment.** This systematic review and meta-analysis study used a risk of bias assessment tool developed by Hoy et al [65] consisting of ten items that assess four domains of bias, internal and external validity. The first four items (items1–4) evaluate the presence of

**Table 1. Descriptions of the studies used in the systematic review and meta-analysis for the knowledge, attitude, and practice of kangaroo mother care among post-natal women in Ethiopia.**

| Author/year | Study region | Study Setting | Study Design | Sample Size | Response rate | Good knowledge | Good attitude | Good practice | GA | KMC Type | Weight (KG) | Study quality |
|---|---|---|---|---|---|---|---|---|---|---|---|---|
| Mose et.al/2021 | SNNP | Hospital | Cross-sectional | 382 | 100 | 35.5 | 50 | 35.3 | Any age | SSC+BF | Any weight | Low risk |
| Gebre et.al/2018 | Somali | Community | Cross-sectional | 829 | 98.3 | Not reported | Not reporter | 23 | Any age | SSC+BF | Any weight | Low Risk |
| Roba AA/2018 | Harar & DireDawa | Hospital | Cross-sectional | 349 | 100 | 69.91 | 63.33 | 54.51 | Any age | SCC only | <2.5kg | Low Risk |
| Jamie, A.H /2020 | Harar | Hospital | Cross-sectional | 166 | 100 | 82.53 | 82.53 | 32.13 | Any age | SSC only | <2.5kg | Low Risk |
| Alelign, Zewuditu (un-pub) | Addis Ababa | Hospital | Cross-sectional | 249 | 100 | 69.1 | 54.22 | 43 | Preterm | SSC+BF | <1.5kg | Low Risk |
| Bedaso et.al/2019 | Amhara, Addis Ababa, Oromia | Hospital | Cross-sectional | 384 | 100 | Not reported | Not Reported | 40.1 | Any age | SSC | Any weight | Low Risk |
| Getinet et.al/2019 | SNNP | Hospital | Cross-sectional | 86 | 92 | 68.6 | 57 | 61.6 | Preterm | SSC only | <2.5kg | Low Risk |
| Dawit, Aster (un-pub) | Addis Ababa | Hospital | Cross-sectional | 297 | 100 | Not reported | Not reporter | 71 | Preterm | SSC+BF | <1.5 | Low Risk |
| Dabere et.al/2020 | National | Community | Cross-sectional | 7488 | Not reported | Not reported | Not reported | 24.3 | Any age | SSC only | Any weight | Low Risk |
| Ebrahim yesuf et.al/2018 | SNNP | Community | Cross-sectional | 215 | 100 | Not reported | Not reporter | 41.9 | Preterm | SSC only | <2.5kg | Low Risk |
| M.W,Ayele et.al/2021 | Amhara | Community | Cross-sectional | 190 | 97 | Not reported | Not reported | 46.8 | Any age | SSC only | <2.5kg | Low Risk |
| Haftey Gebremedihn et,al (un-pub) | Tigray | Hospital | Cross-sectional | 397 | 96.6 | Not reported | Not reported | 54.4 | Any age | SSC only | <1.5kg | Low Risk |
| Lakew W. and B. Worku/2014 | Addis Ababa | Hospital | Cross-sectional | 110 | Not reported | Not reported | Not reported | 83 | Preterm | SSC only | <1.5kg | Low Risk |
| Weldeargay et.al/2019 | National | Hospital | Cross-sectional | 768 | Not reported | Not reported | Not reporter | 46.4 | Preterm | SSC only | <1.5kg | Low Risk |
| Demissie et.al/2018 | Addis Ababa | Hospital | Cross-sectional | 356 | 100 | Not reported | Not reporter | 47.2 | Preterm | SSC+BF | <1.5kg | Low Risk |
| Emishaw et.al/ | Tigray | Hospital | Cross-sectional | 109 | Not reported | Not reported | Not reporter | 28.12 | Preterm | SSC only | <2.5kg | Low Risk |

selection bias, non- response bias and external validity. The other six items (items 5–10) assess the presence measuring the bias, analysis- related bias and internal validity. Therefore, if studies that received 'yes' for eight or more of the ten questions were classified as 'low risk of bias.' If studies that received 'yes' for six to seven of the ten questions were classified as 'moderate risk' whereas if studies that received 'yes' for five or fewer of the ten questions were classified as 'high risk' as reported in (S3 File).

## Statistical analysis

After data extraction was done using Microsoft Excel, the analysis was conducted by using STATA version 14 statistical software. Publication bias was checked by funnel plot and more objectively through Begg and Egger's regression tests, with P< 0.05 considered to indicate potential publication bias. A trim and fill analysis was done to see the effect of publication bias. It adds studies to form the symmetrical distribution. The presence of between-study

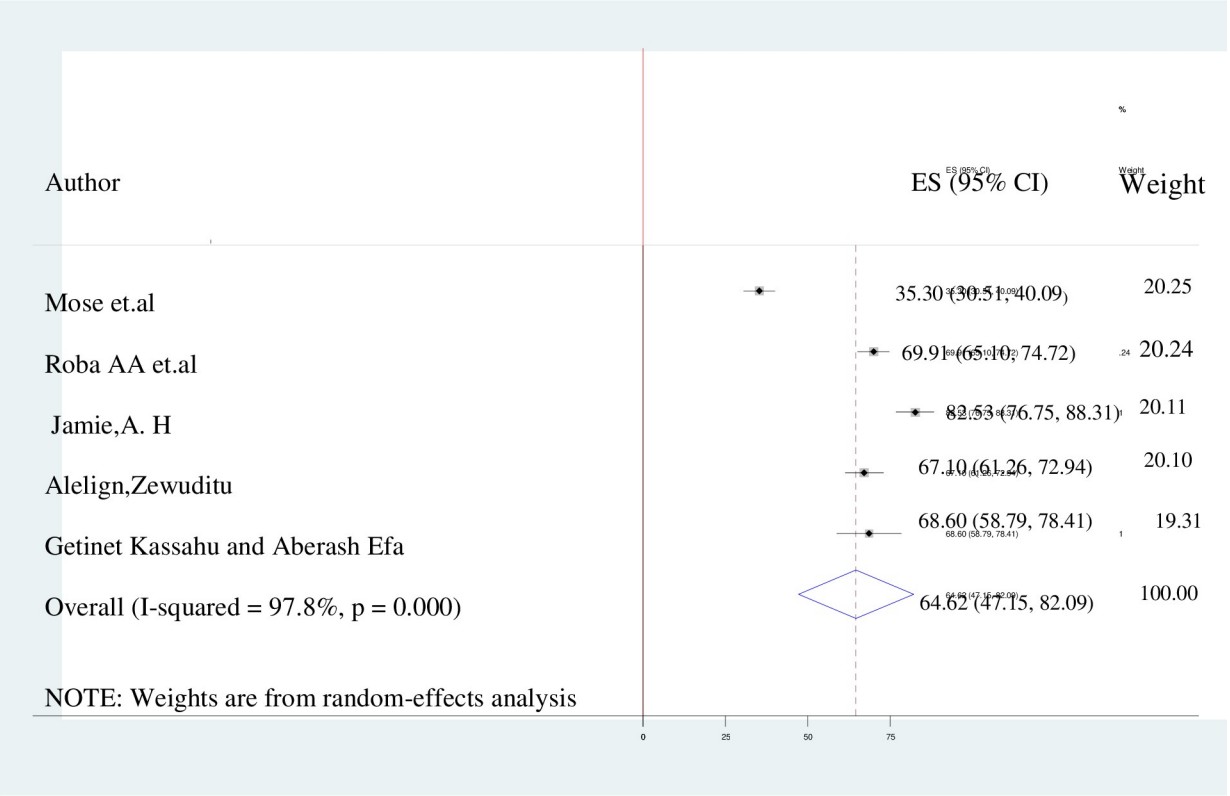

**Fig 2. Forest plot of knowledge with the height of the diamond is the overall effect size (64.62% while the width is the confidence interval at 95% (47.15%–82.09%).** The y-axis shows the standard error of each study while the x-axis the estimate of effect size of the each study. The vertical line denotes the no effect. The box represents the effect size of each study and the line across the box is confidence interval of each study.

heterogeneity was checked by using Cochrane Q statistic. This heterogeneity between studies was quantified using $I^2$, in which a values of 0, 25, 50, and 75% represented no, low, medium, and in-creased heterogeneity, respectively. A forest plot was used to visually assess the presence of heterogeneity, which presented at a high level random-effect model was used for analysis to estimate the overall prevalence of knowledge, attitude and practice of kangaroo mother care. Subgroup analysis was done by study setting, region, gestational age, birth weight, and type of kangaroo care practice. A sensitivity analysis was executed to see the effect of a single study on the overall prevalence of the meta-analysis estimate. The findings of the study were presented in the form of text, tables, and figures.

## Results

### Study selection

There were 273 research articles retrieved using an electronic search. Of these articles, 117 were expelled for duplication and 95 studies were excluded after reviewing their titles and abstracts. At the qualification stage, 61 articles were completely gotten to and evaluated for the capability. Finally, 16 studies [48–63] with 12,475 participants were included in this systematic review and meta-analysis. All studies were cross-sectional, and reported the prevalence of knowledge, attitude, and practice of kangaroo mother care **(Fig 1)**.

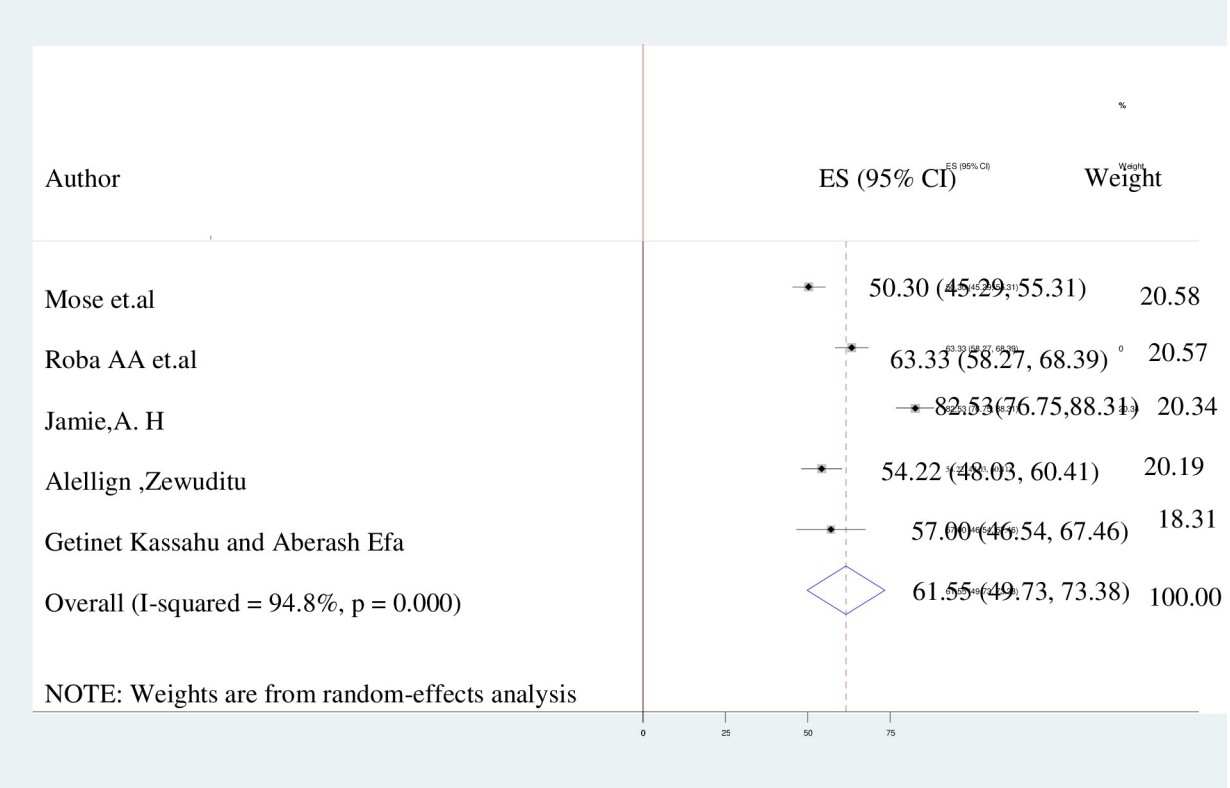

**Fig 3. The forest plot of attitude with the diamond represents the summary point estimate (61.55%) and the horizontal extremity of the diamond is the confidence interval at 95% (49.73%–73.38).** The standard error is plotted at the y-axis and the effect size plotted at x-axis. The squares represent the effect estimate of the individual studies and the horizontal lines indicate the confidence interval; the dimension of the square reflects the weight of each study.

### Description of included studies

Out of 273 articles retrieved at first, sixteen articles met the eligibility criteria and were included in the final meta-analysis as reported by Fig 1. The author's names, publication year, study design, sample size, study region, study setting, response rate, birth weight, kangaroo mother care type, gestational age, the prevalence of knowledge, attitude, and practice of kangaroo mother care listed in the below Table 1.

Four studies were found in Addis Ababa [49, 53–55], three in Southern Nations Nationalities and Peoples Region [48, 50, 60], two at the national level [56, 58], one in Amhara [59], two in Tigray [57, 63], one in Somali [61], one in Harar and Dire Dawa [51], one in Harar [52], and one in Addis Ababa, Amhara, Oromia and Benishanguel Gumuize [62]. Of the sixteen cross-sectional studies, twelve were institutional-based, and four were community-based. The earliest in 2014 and the latest in 2021. The sample sizes ranged from 86 to 7488. The prevalence of knowledge, attitude, and practice of kangaroo mother care ranged were ranged from 35.5%-82.53%, 50%-82.53%, and 28%-83% respectively. The response rate ranged from 92 to 100 percent. Eight studies reported any gestational age (preterm, term, and post-term) infants, while the remaining only on preterm infants. All sixteen studies were assessed by using Joanna Briggs Institute (JBI) quality appraisal checklist. All of these studies had reported a low risk (**Table 1**).

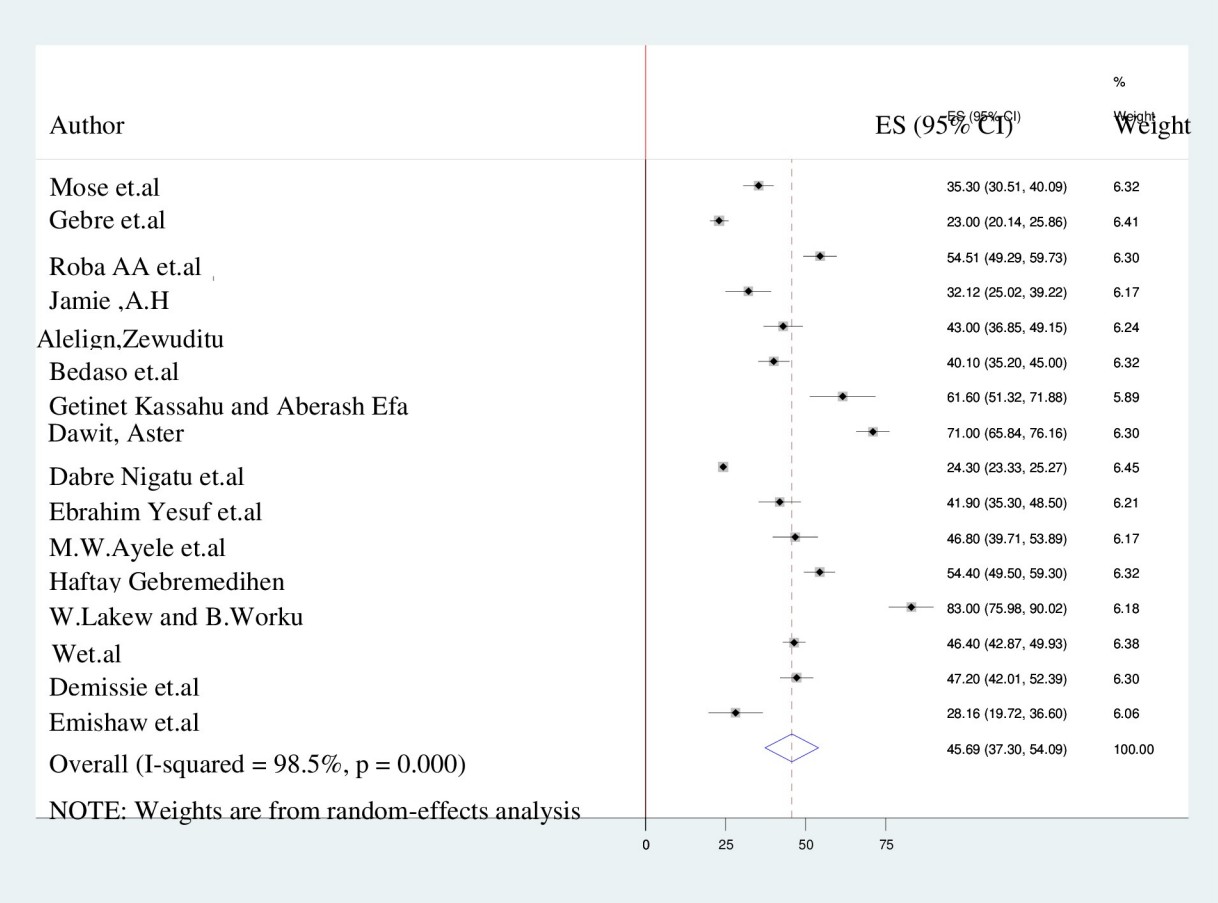

**Fig 4. The forest plot of practice with the height of the diamond is the overall effect size (45.7%) while the width is the confidence interval at 95% (37.30%-54.09%).** The y-axis shows the standard error of each study while the x-axis the estimate of effect size of the each study. The vertical line denotes the no effect. The square represents the effect size of each study and the line across the square is confidence interval of each study.

## Level of knowledge, attitude, and practice towards kangaroo mother care

The pooled prevalence of the knowledge about, attitudes towards and practice of kangaroo mother care in Ethiopia is presented by the forest plots in **Figs 2–4**. A random-effect model showed that the pooled level of good knowledge was 64.62% (95% CI: 47.15%–82.09%; $I^2$ = 97.8%). The overall estimated positive attitude towards kangaroo mother care was 61.55% (49.73%–73.38%; $I^2$ = 94.8%), while the pooled estimate of poor practice of kangaroo mother care among postnatal women was 45.7% (95% CI: 37.30%–54.09%; $I^2$ = 98.5%.

## Leave–one- out sensitivity analysis

A leave-one-out sensitivity analysis was carried out to detect the effect of each study on the overall prevalence of a good level of knowledge about, a positive attitude towards and a poor level of practice of kangaroo mother care among postnatal women by excluding one study at a time. The results showed that the excluded study leads to significant change in the overall prevalence of a good level of knowledge, positive attitude and poor practice. In the sensitivity analysis, both Jamie,AH. and Mose et al. showed an impact on the pooled level of good knowledge

**Table 2. A leave–out-one sensitivity analysis for knowledge, attitude, and practice of kangaroo mother care among postnatal women in Ethiopia.**

| Knowledge related articles | | |
|---|---|---|
| Study omitted | Pooled estimate | 95%CI |
| Mose et.al | 72.22 | 64.92–79.53 |
| Roba AA et.al. | 63.31 | 40.30–86.32 |
| Jamie, A.H | 60.11 | 41.44–78.77 |
| Zewuditu Alelign | 64.02 | 41.59–86.45 |
| Getinet et.al | 63.67 | 43.23–84.12 |
| Attitude related articles | | |
| Study omitted | Pooled estimate | 95%CI |
| Mose et.al | 64.48 | 51.46–77.50 |
| Roba AA et.al. | 61.07 | 44.97–77.17 |
| Jamie, A.H. | 56.17 | 49.59–62.76 |
| Alelign,zewuditu | 63.39 | 48.82–77.96 |
| Getinet et.al | 62.58 | 48.82–76.33 |
| Practice related articles | | |
| Study omitted | Pooled estimate | 95%CI |
| Mose.et.al | 46.40 | 37.44–55.36 |
| Roba AA et.al | 45.10 | 36.52–53.68 |
| M.W. Ayele et.al | 45.62 | 36.90–54.35 |
| Getinet et.al | 44.70 | 36.12–53.28 |
| Gebere et.al | 47.26 | 37.83–56.69 |
| Dawit, Aster | 43.96 | 36.11–51.82 |
| Bedaso et.al | 46.08 | 37.17–54.98 |
| Dabere et.al | 47.16 | 38.94–55.38 |
| Demissie et.al | 45.60 | 36.83–54.36 |
| Ebrahim Yesuf | 45.95 | 37.17–54.73 |
| Emishaw et.al | 46.83 | 38.07–55.59 |
| Jamie,A.H. | 46.59 | 37.79–55.40 |
| Zewditu Alelign | 45.88 | 37.09–54.67 |
| Tesfaye Geberemedihn | 45.10 | 36.54–53.67 |
| Weldearagay et.al | 45.65 | 36.75–54.55 |
| Lakew. W and B. Worku | 43.22 | 35.37–51.07 |

and positive attitude, while Lakew B, B. Worku and Gebre et al. showed an impact on the level of poor practice of kangaroo mother care (**Table 2**).

## Subgroup analysis

The subgroup analysis based on kangaroo mother care component showed that a level of good knowledge was 74% in only skin-to-skin contact and 51.2% in skin-to-skin contact with exclusive breastfeeding. The level of positive attitude towards kangaroo mother care component was 67.98% in skin-to-skin contact only and 51.85% for skin-to-skin contact with exclusive breastfeeding. In this subgroup analysis, the level of poor practice was examined by study region, study setting, publication, component of kangaroo mother care, gestational age and birth weight. The pooled level of poor kangaroo mother care practice by region was 60.99% in Addis Ababa and 39.92% in another region (where a study conducted in single region/multiple regions). In the case of kangaroo mother care

**Table 3. The overall estimated level of good knowledge, positive attitude, and poor practice towards kangaroo mother care in Ethiopia, 95%CI and heterogeneity estimate with a p-value for sub-group analysis.**

| Knowledge related articles | | | |
|---|---|---|---|
| Variable | Characteristics | Pooled level of good knowledge 95%(CI) | $I^2$(p-value) |
| Kangaroo mother care type | SSC only | 74% (64.744–83.236) | 83.1%(0.003) |
| | SSC+BF | 51.2%(19.991–82.318) | 98.4% (0.000) |
| Attitude related articles | | | |
| Variable | Characteristics | Pooled level of positive Attitude 95%(CI) | $I^2$(p-value) |
| Kangaroo mother care type | SSC only | 67.98%(52.968–82.997) | 93.5%(0.000) |
| | SSC+BF | 51.85%(47.958–55.749) | 0.0%(0.335) |
| Practice related articles | | | |
| Variables | Characteristics | Pooled level of poor practice 95%(CI) | $I^2$(p-value) |
| Study setting | Community | 33.067% (25.397–40.738) | 95.4%(0.000) |
| | Institutional | 49.697%(41.801–57.594) | 95.9% (0.000) |
| Region | Addis Ababa | 60.992%(43.326–78.658) | 97.3%(0.000) |
| | SNNP | 45.505%(32.660–58.349) | 90.4%(0.000) |
| | Tigray | 45.694%(37.297–54.092) | 96.4%(0.002) |
| | Other | 39.228%(26.286–52.170) | 97%(0.002) |
| | Nationwide | 35.282%(13.625–56.939) | 99.3% (0.001) |
| Publication | Published | 43.217% (34.943–51.492) | 98.2%(0.000) |
| | Unpublished | 56.200%(40.808–71.591) | 95.9%(0.000) |
| Kangaroo mother care type | SSC only | 46.572%(35.565–57.579) | 98.6%(0.000) |
| | SSC+BF | 43.844% (26.763–60.925) | 98.6%(0.000) |
| Gestational age | Preterm | 52.774%(41.867–63.681) | 96.3%(0.000) |
| | Any age | 38.675%(29.920–47.429) | 97.9%(0.000) |
| Birth weight | <1.5kg | 57.372%(46.223–68.522) | 96.5%(0.000) |
| | 1.5–2.5kg | 44.080% (34.683–53.478) | 90.3%(0.000) |
| | Any weight | 30.274%(23.871–36.677) | 94.8%(0.000) |

components, the prevalence of poor practice was 46.57% in only skin-to-skin contact practice and 43.84% in skin-to-skin contact with exclusive breastfeeding. Regarding gestational age, the level of poor practice was found to be 52.77% in preterm neonates and 38.68% at any gestational age. The level of poor practice relative to birth weight was 57.37% in birth weight below than 1.5 kg and 30.27% at any weight (**Table 3**).

## Publication bias

The presence of publication bias was checked using funnel plot visualisation and Egger's and Begg's regression tests (P<0.05). The Egger and Begg tests both revealed no statistical evidence of publication bias for a good level of knowledge (P = 0.577 and P = 0.240, respectively; **Fig 5**). There was also no statistical evidence of publication bias for a positive attitude in terms of the Egger (P = 0.928) and Begg (p = 0.624) tests (**Fig 6**). The results of the Begg (P = 0.000) and Egger (P = 0.000) tests show the presence of publication bias for the level of poor practice of kangaroo mother care. Additionally, an asymmetric distribution was visualised on the funnel plot (**Fig 7**). Subsequently, trim-and-fill analysis was performed and indicated the presence of seven unpublished studies (**Fig 8**). A counter-enhanced funnel plot was also calculated, and the missing studies in the areas of higher statically significance suggested that the cause of asymmetry was due to factors other than publication bias, such as the study variables (**Fig 9**).

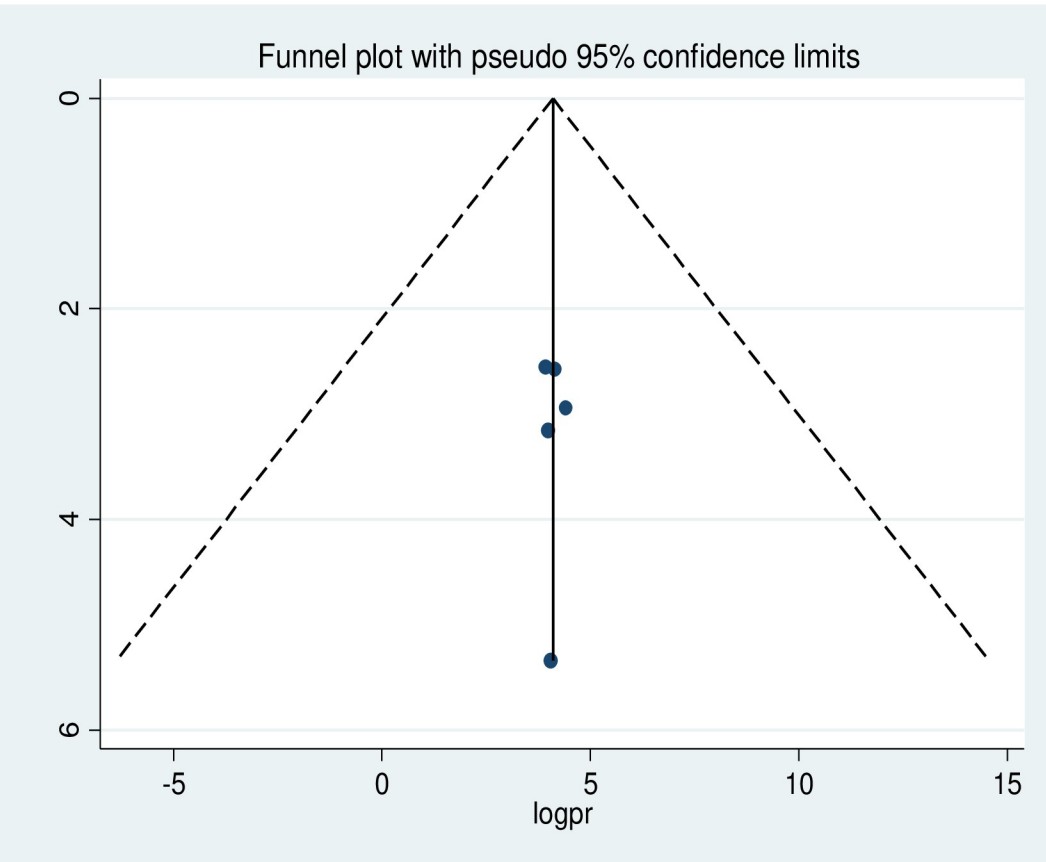

**Fig 5. Funnel plot showing symmetrical distribution of studies indicating absence of publication bias.** The Y-axis is the standard error and the X-axis is the study result or effect size. The dotted diagonal lie of the funnel is the 95% confidence interval and the vertical. The vertical line is the line of no-effect and dots are included studies reporting knowledge of kangaroo mother care.

## Discussion

Kangaroo mother care was first introduced in Ethiopia in 1996 at the Black Lion Hospital in Addis Ababa. Since then, kangaroo mother care services have expanded to other hospitals and health facilities at all levels. Nowadays, they have been issued in a series of policy documents by the Federal Ministry of Health, the New born and Child Survival Strategy 2015–2020, the Health Sector Transformation Plan, and the National Healthcare Quality Strategy. Despite a governmental emphasis on reducing neonatal mortality via evidence-based strategies such as kangaroo mother care, Ethiopia is one of the sub-Saharan African countries with the most neonatal mortalities.

Due to the scant availability of literature, we conducted this systematic review and meta-analysis to better understand the knowledge, attitudes and practices among Ethiopian women. We included all available studies using a variety of electronic search engines and were also able to undertake a sub-group analysis assessing the proportion of knowledge, attitudes and practices by study setting, study region, gestational age, component of kangaroo mother care, birth weight and publications.

This study's findings reveal that the pooled prevalence of good kangaroo mother care knowledge was 64.62%. This is lower than the results of a study conducted in the USA (75%) among nurses working in intensive care units [66]. This inconsistency can be attributable to

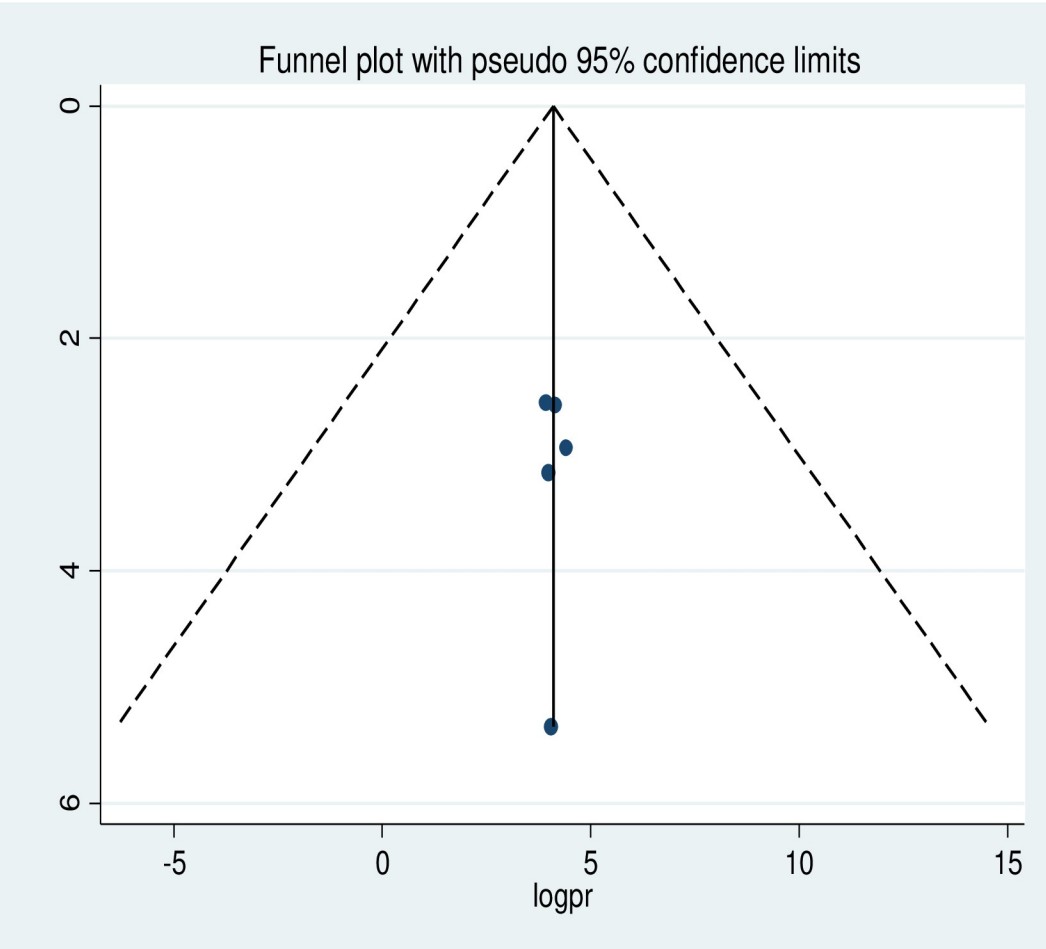

**Fig 6. Funnel plot showing symmetrical distribution of studies indicating absence of publication bias.** The Y-axis is the standard error and the X-axis is the study result or effect size. The dotted diagonal lie of the funnel is the 95% confidence interval and the vertical. The vertical line is the line of no-effect and dots are included studies reporting attitude of kangaroo mother care.

existing socio-economic, healthcare system infrastructure, study population variation, and methodological differences (including study design) across the studies. The sub-group analysis, based on components of kangaroo mother care, showed that participants were more knowledgeable concerning skin-to-skin contact only compared to both skin-to-skin contact and exclusive breastfeeding. This finding is not surprising because exclusive breastfeeding has received much attention lately.

More than half (61.5%) % of women who gave birth in Ethiopia had a positive attitude towards kangaroo mother care. This finding is consistent with a US result, which showed that over 60% of respondents had a positive perception of kangaroo mother care [66]. The sub-group analysis also revealed that participants had a positive attitude towards skin-to-skin contact only over both skin-to-skin contact and exclusive breastfeeding, which might be due to the lack of long-term exposure to exclusive breastfeeding.

The pooled level of poor kangaroo mother care practice was 45.7%. This was relatively lower than the findings of studies conducted in New Zealand (69%) and Singapore (92%) [67, 68]. One conceivable clarification for the disparity is the methodological differences, such as

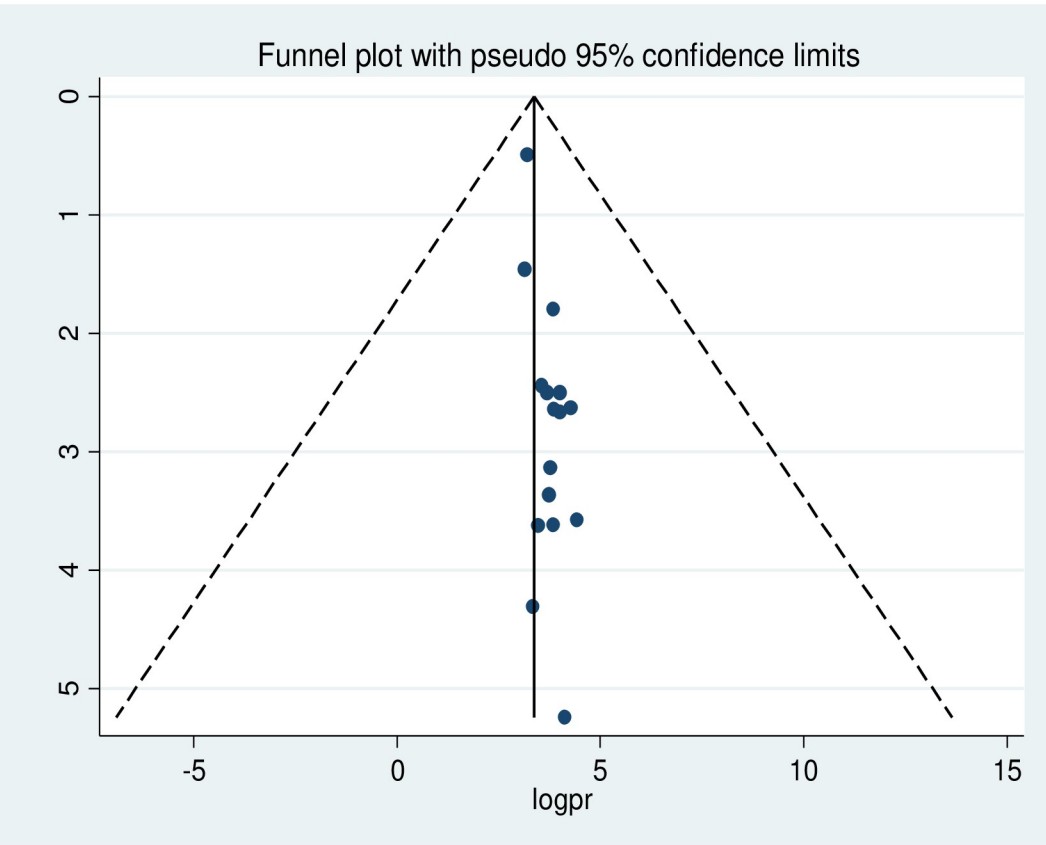

**Fig 7. Funnel plot showing asymmetrical distribution of studies indicating the presence of publication bias.** The Y-axis is the standard error and the X-axis is the study result or effect size. The dotted diagonal line of the funnel is the 95% confidence interval and the vertical. The vertical line is the line of no-effect and dots are included studies reporting attitude of kangaroo mother care.

study design and socio-cultural difference, between the countries, which, in turn, affect the practice of kangaroo mother care. Furthermore, Ethiopia is a third-world country with a poor economic status and poor maternal and paediatric health coverage, along with low levels of skilled deliveries. The sub-group analysis showed that poor practices were observed in the community-based study, skin-to-skin contact and exclusive breastfeeding (at any weight or gestational age); in contrast, a high level of practice was reported in Addis Ababa. This indicates the significance of kangaroo mother care for very-low-birth-weight, low-birth-weight and preterm neonates, as well as the need to enhance the procedure at the community level. Regarding region, Addis Ababa had a greater share of kangaroo mother care practice, which might be due to the adaptation of the procedure by both mothers and healthcare providers for a long time since its introduction at the Black Lion Hospital (one of Ethiopia's earliest specialised hospitals), located in Addis Ababa, Ethiopia.

In this study, we utilised a random-effect model to manage a significant variation that resulted in between-study heterogeneity. We assessed leave-one-out sensitivity and the results show that every study had a significant impact on the pooled good level of knowledge, positive attitude, and poor level of kangaroo mother care practice. We also assessed the possible variability source via sub-group analysis using the study settings, regions, publications, gestational ages, birth weights

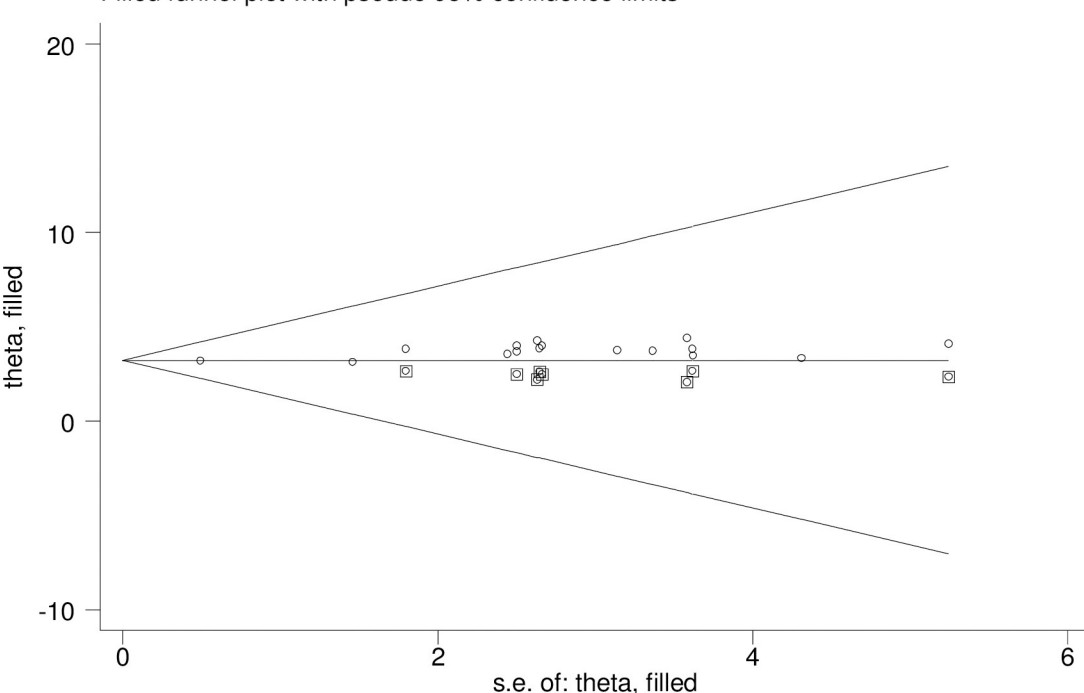

**Fig 8. The funnel plot for trim-and-fill method was used to correct the result seven potential missing studies were required in the left side of the funnel plot to ensure symmetry.** The enclosed circles represent the dummy studies and the free circles are genuine studies.

and components of kangaroo mother care. The high heterogeneity might be due to differences in the sample populations, paper qualities, or socio-cultural, ethnic and regional differences.

## Conclusion

In conclusion, this systematic review and meta-analysis reported that there was a significant gap in knowledge, attitudes, and practices of kangaroo mother care among Ethiopian women. Besides, the pooled prevalence of knowledge, attitudes, and practices differed based on the study settings, regions, publications, gestational ages, birth weights and components of kangaroo mother care. The information generated from these findings should be used for the provision of accurate and up-to-date training and education of kangaroo mother care. Accordingly, it is better to have periodic kangaroo mother care training for postnatal women and, subsequently, expand service dimensions across the community level.

### Strength and limitation of the study

The strength of the study including the use of a comprehensive electronic search strategy through the variety of datasets to determine the overall level of knowledge, attitude, and practice of kangaroo mother care, the use of JBI-MAStARI appraisal, and the accessing of grey literature's. This study also had some limitations. These were the absence of a standard definition of good knowledge, positive attitude, and poor practice of kangaroo mother care to operationalize by the research team and might be researched bias on a cut of point. The absence of a similar previous study makes it is very difficult to compare the

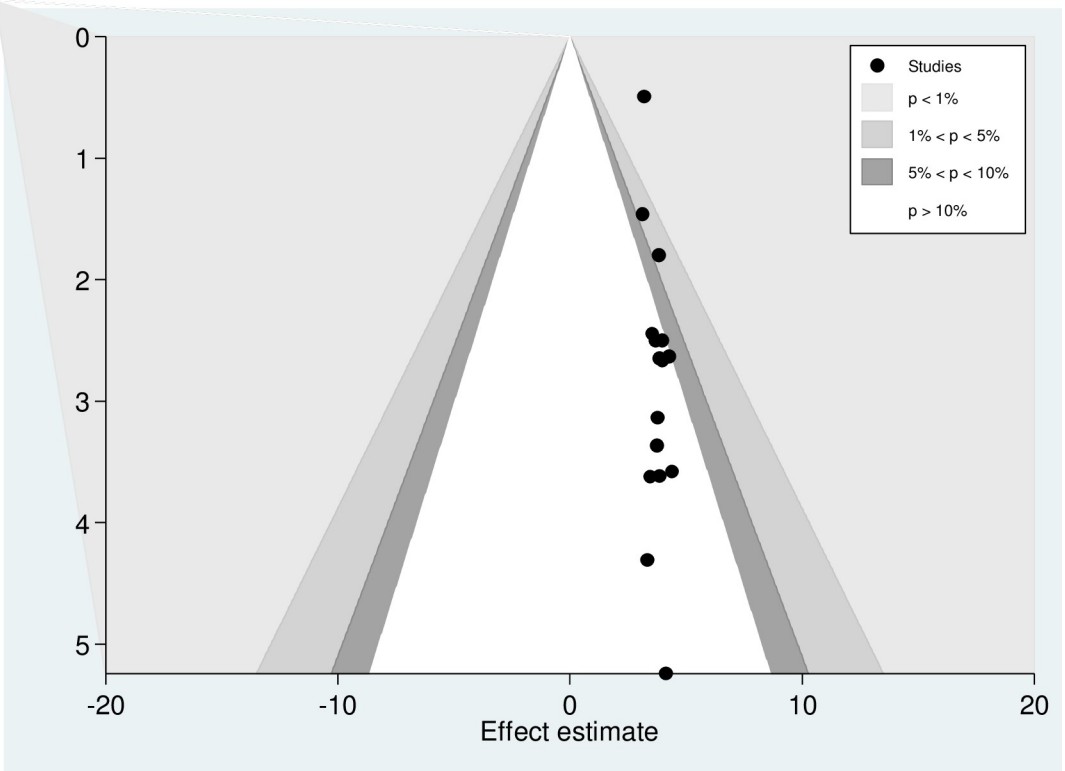

**Fig 9. Counter-enhanced funnel plot suggestions of missing studies on the bottom left-hand-side of the plot.** Since the majority of this area contains regions of high statistical significance ($P < 0.01$), this reduces the plausibility that publication bias is the underlying cause of this funnel asymmetry. Various shaded regions indicate statistical significance. In particular, the white shaded region in the middle corresponds to p-values greater than .10, the dark gray-shaded region corresponds to p-values between .10 and .05, the medium gray-shaded region corresponds to p-values between .05 and .01, and the region outside of the funnel corresponds to p-values below .01.

findings of this study. The study had included the whole regions of the country, but Afar and Gambela.

## Supporting information

**S1 File. Prisma checklist.**
(PDF)

**S2 File. Methodological quality assessment of included studies using Joanna Brigg's Institute quality appraisal criteria scale (JBI).** The eight item questions assessing inclusion criteria, study setting and participant, exposure measurement, objectives, confounder, statically analysis, outcome measurement and dealing confounder were used.
(PDF)

**S3 File. Risk of bias assessment for the included studies.** The ten item questions of which four items assess external and six items assess internal validity were used.
(PDF)

## Author Contributions

**Conceptualization:** Natnael Atnafu Gebeyehu, Kelemu Abebe Gelaw.

**Data curation:** Natnael Atnafu Gebeyehu.

**Formal analysis:** Natnael Atnafu Gebeyehu, Kelemu Abebe Gelaw, Gedion Asnake Azeze, Biruk Adie Admass.

**Funding acquisition:** Kelemu Abebe Gelaw.

**Investigation:** Natnael Atnafu Gebeyehu, Gedion Asnake Azeze, Getachew Asmare Adela.

**Methodology:** Natnael Atnafu Gebeyehu, Kelemu Abebe Gelaw, Gedion Asnake Azeze, Biruk Adie Admass, Eyasu Alem Lake, Getachew Asmare Adela.

**Resources:** Biruk Adie Admass, Eyasu Alem Lake.

**Software:** Natnael Atnafu Gebeyehu, Kelemu Abebe Gelaw, Gedion Asnake Azeze, Eyasu Alem Lake, Getachew Asmare Adela.

**Supervision:** Kelemu Abebe Gelaw, Gedion Asnake Azeze, Eyasu Alem Lake, Getachew Asmare Adela.

**Validation:** Natnael Atnafu Gebeyehu, Gedion Asnake Azeze, Eyasu Alem Lake.

**Visualization:** Biruk Adie Admass.

**Writing – original draft:** Natnael Atnafu Gebeyehu, Gedion Asnake Azeze.

**Writing – review & editing:** Natnael Atnafu Gebeyehu, Kelemu Abebe Gelaw, Biruk Adie Admass, Eyasu Alem Lake.

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
