## [Decision Letter · Decision Letter 0]

13 Oct 2021

PONE-D-21-27069Knowledge, Attitude and Practice towards kangaroo mother care among postnatal women in Ethiopia: Systematic review and Meta-analysisPLOS ONE

Dear Dr. Natnael Atnafu

Thank you for submitting your manuscript to PLOS ONE. After careful consideration, we feel that it has merit but does not fully meet PLOS ONE’s publication criteria as it currently stands. Therefore, we invite you to submit a revised version of the manuscript that addresses the points raised during the review process.

ACADEMIC EDITOR: 

There are several typological and grammar usage errors that need extensive proof reading for revisions. This would help you increase the readability of the manuscript.

**Methods **

Explain the reliability and validity of the data extraction tool. Explain if data transformation was required or undertaken when data were reported differently on the considered factors and/outcome of interest. 

We look forward to receiving your revised manuscript.

Kind regards,

Wubet Alebachew Bayih, M.Sc.

Academic Editor

PLOS ONE

Journal Requirements:

Whilst you may use any professional scientific editing service of your choice, PLOS has partnered with both American Journal Experts (AJE) and Editage to provide discounted services to PLOS authors. Both organizations have experience helping authors meet PLOS guidelines and can provide language editing, translation, manuscript formatting, and figure formatting to ensure your manuscript meets our submission guidelines. To take advantage of our partnership with AJE, visit the AJE website (http://aje.com/go/plos) for a 15% discount off AJE services. To take advantage of our partnership with Editage, visit the Editage website (www.editage.com) and enter referral code PLOSEDIT for a 15% discount off Editage services.  If the PLOS editorial team finds any language issues in text that either AJE or Editage has edited, the service provider will re-edit the text for free.

3. Please confirm that you have included all items recommended in the PRISMA checklist including:

- The dates used for the literature search, and when this search was carried out.

- A Supplemental file of the results of the individual components of the quality assessment, not just the overall score, for each study included.

- Please see http://www.prisma-statement.org/PRISMAStatement/PRISMAEandE for guidance on reporting.

“Not applicable”

5. We note that you have referenced

“Alelign, Zewuditu/unpublished

Dawit , Aster /unpublished

Haftey Gebremedihn et,al /unpublished” which has currently not yet been accepted for publication. Please remove this from your References and amend this to state in the body of your manuscript: (ie ““Alelign, Zewuditu [unpublished]”) as detailed online in our guide for authors

Reviewers' comments:

Reviewer's Responses to Questions

**Comments to the Author**

1. Is the manuscript technically sound, and do the data support the conclusions?

Reviewer #1: Yes

Reviewer #2: Yes

2. Has the statistical analysis been performed appropriately and rigorously? 

Reviewer #1: Yes

Reviewer #2: Yes

3. Have the authors made all data underlying the findings in their manuscript fully available?

Reviewer #1: Yes

Reviewer #2: No

4. Is the manuscript presented in an intelligible fashion and written in standard English?

Reviewer #1: Yes

Reviewer #2: Yes

5. Review Comments to the Author

Reviewer #1: This is an interesting study even though in Ethiopia the KMC was introduced in 1996. There are a lot of publication regarding KMC. So the author team has the opportunity to do systematic review and meta analysis on the issue.

Method:

I would like to suggest the time of data collection and range of yearly published on the articles to be reviewed.

The search strategy in the Pub Med : ((((( Knowledge, please check is it typo error?

Please in the keyword it included word: Ephysiotomy I could not find the relation with the subject to be studied.

Operational definition:

Practice was the study participants who responded more than 50% responded questions were categorized as having a good ATTITUDE (???). Please check.

Eligibility criteria:

I would like to recommend to add the time frame and also include grey materials (please explain what does it mean).

Data analysis:

Any ideal difference that happened due to extraction were also selected by the fourth articles. Please explain what you meant is by the four persons research team member?

Reviewer #2: Reviewer Comments to Authors:

Thank you for the opportunity to review your valuable research.

1. Abstract:

Since this is a Systematic Review (SR) article, it is advisable to follow the guideline in reporting a SR as listed in PRISMA Guideline. It seems that the abstract did not contain information about inclusion and exclusion criteria, methods to assess risk of bias, a brief summary of limitations, register name and registration number. It would be more complete if the abstract could contain the above information that are lacking.

2. Introduction:

The introduction was well-written.

3. Methods:

The methods section was, in general, written well. Some missing information could be added:

a. The search strategy was listed only for Pubmed, but not mentioned for the other databases. Could that be also provided in the article?

b. For Selection Process and Data Collection Process, if automation tool was not used, please add information as such.

c. The use of grey literature was not mentioned in the method section, however, it was mentioned in section on Strength and Limitation of the Study (third line) that one the strength of this study is …”the accessing of grey literature’s”. Could you explain this?

4. Results:

The results section was also well-written, however, there is some a question:

a. Sensitivity analysis, it is written as such: “A leave-out-one sensitivity analysis was done to identify the effect of each study on the pooled prevalence of good level of knowledge, positive level of attitude, and poor level of practice of kangaroo mother care of postnatal women by excluding each study step by step. The result showed that the excluded study brings significant change to the overall prevalence of a good level of knowledge, positive attitude, and poor practice.” Based on Table 1, Good Knowledge was supported by 5 studies, Good Attitude by 5 studies, and Good Practice by 16 studies, and from the Forest Plots (Fig. 2, Fig. 3, and Fig. 4), the number of studies listed in Table 1 and the corresponding figures of forest plot were exactly the same. There was no exclusion of any studies. The facts that there were only 5 studies for Good Knowledge and Good Attitude, is because the number of studies that provided information for the two outcomes were only five. There was no exclusion of any study as claimed above. Could the authors clarify this matter?

b. How to explain the connection between data estimates listed on Table 2 and those in figures of forest plot (Figs. 2, 3, and 4).

5. Discussion:

In the discussion section, the findings from this SR was compared to findings from national surveys done in developed countries (Denmark, America – meaning USA?, Taiwan, and Canada). Would the comparison be comparable? First, national survey data form the mentioned developed countries usually includes large sample sizes, whereas the studies included in this SR involved relatively small number of respondents. Secondly, eleven of 16 studies were from hospitals, whereas national surveys would include respondents from the community. Thirdly, would comparing KMC provided by mothers from Ethiopia to those from the above mentioned countries be a suitable comparisons? Especially if we consider the differences based on social, economic, cultural and daily needs/problems faced by the mothers. It would be more meaningful to compare the findings with those from countries in Africa or developing countries in Asia, as well as in Latin America.

Thank you. I hope that this Systematic Review will add significant knowledge about KMC implementation.

6. PLOS authors have the option to publish the peer review history of their article (what does this mean?). If published, this will include your full peer review and any attached files.

Reviewer #1: **Yes: **Hadi Pratomo

Reviewer #2: No

---

## [Author Response · Author response to Decision Letter 0]

18 Feb 2022

The topological and grammatical problems as well as plos one format style of raised by the editor was revised deeply as presented in revised manuscript document. The same is true for the comments raised by the reviewers were revised deeply as much as we can.

---

## [Editor Report · Decision Letter 1]

2 Mar 2022

Knowledge, attitude and practice towards kangaroo mother care among postnatal women in Ethiopia: systematic review and meta-analysis

PONE-D-21-27069R1

Dear Dr. Atnafu,

We’re pleased to inform you that your manuscript has been judged scientifically suitable for publication and will be formally accepted for publication once it meets all outstanding technical requirements.

Kind regards,

Wubet Alebachew Bayih, M.Sc.

Academic Editor

PLOS ONE
---

## [Editor Report · Acceptance letter]

10 Mar 2022

PONE-D-21-27069R1 

Knowledge, attitude and practice towards kangaroo mother care among postnatal women in Ethiopia: systematic review and meta-analysis 

Dear Dr. Gebeyehu:

I'm pleased to inform you that your manuscript has been deemed suitable for publication in PLOS ONE. Congratulations! Your manuscript is now with our production department. 

Kind regards, 

on behalf of

Dr. Wubet Alebachew Bayih 

Academic Editor

PLOS ONE